# Standardized Computer-Assisted Analysis of PRAME Immunoreactivity in Dysplastic Nevi and Superficial Spreading Melanomas

**DOI:** 10.3390/ijms24076388

**Published:** 2023-03-28

**Authors:** Elias A. T. Koch, Michael Erdmann, Carola Berking, Franklin Kiesewetter, Rafaela Kramer, Stefan Schliep, Markus V. Heppt

**Affiliations:** 1Department of Dermatology, Uniklinikum Erlangen, Friedrich-Alexander-University Erlangen-Nürnberg (FAU), 91054 Erlangen, Germany; 2Comprehensive Cancer Center Erlangen-European Metropolitan Area of Nuremberg (CCC ER-EMN), 91054 Erlangen, Germany; 3MVZ Pathology, Sozialstifung Bamberg, 96049 Bamberg, Germany

**Keywords:** PRAME, melanoma, dysplastic nevus, immunohistochemistry, computer-assisted analysis

## Abstract

PRAME (PReferentially expressed Antigen in MElanoma) is a cancer testis antigen that is frequently expressed in melanoma compared to benign melanocytic proliferations and nevi. However, the interpretation of the intensity and distribution of PRAME immunostaining is not standardized a lot, which makes interpretation difficult. PRAME-stained histological slides of superficial spreading melanomas (SSM) and dysplastic nevi (DN) were digitized and analyzed using the digital pathology and image platform QuPath. *t*-tests and ROC AUCs were performed with SPSS. A *p*-value of <0.05 was used for statistical significance, and a ROC AUC score of >0.8 was considered a good result. A cut-off score was defined in an evaluation cohort and subsequently analyzed in an independent validation cohort. In total, 81 PRAME-stained specimens were included. The evaluation cohort included 32 (50%) SSM and 32 (50%) DN, and the mean of PRAME-positive cells/mm^2^ for the entire lesion was 455.3 (SD 428.2) in SSM and 60.5 (SD 130.1; *p* < 0.001) in DN. The ROC AUC of PRAME-positive cells of the entire lesion was 0.866, and in the epidermis it was 0.901. The defined cut-off score to distinguish between DN and SSM was 97.67 cells/mm^2^. In the validation cohort, 16 out of 17 cases (94.1%) were correctly classified by the cut-off score. The computer-aided assessment of PRAME immunostaining is a useful tool in dermatopathology to distinguish between DN and SSM. Lesions with a moderate expression and indifferent morphologic features will remain a challenge for dermatopathologists.

## 1. Introduction

For decades, the distinction between dysplastic nevi and melanoma has remained a challenge faced by virtually all practicing dermatopathologists. Histological diagnosis is primarily made with the conventional hematoxylin and eosin (HE) stain through an assessment of the morphological features of the lesion, including pagetoid spread, cytological atypia, dermal mitoses, asymmetry, lack of circumscription, impaired maturation and hypercellularity [1]. Nevertheless, in over 50% of cases, immunohistochemistry (IHC) markers are used to facilitate the diagnosis [2], and numerous studies have focused attention on molecules that can be used in differential diagnoses [1]. Melan-A (also MART-1) and Sry-related HMg-Box gene 10 (SOX-10) are commonly used to stain all melanocytic cells [2], both of which are sensitive and specific markers for melanocytic lesions [3]. Further IHC markers have been established to aid in the differential diagnosis. Notably, 5-hydroxymethylcytosine (5-hmC) shows an intense immunoreactivity in benign nevus cells, whereas it is increasingly lost in dysplastic melanocytic nevi (DN) and frequently absent in melanomas [4,5]. Additionally, p16 shows a decreased nuclear staining within melanomas compared with nevi [6]. On the other hand, expressions of preferentially expressed antigen in melanoma (PRAME) and p53 (also helpful markers for desmoplastic melanoma) are significantly higher in melanomas than in nevi [7]. Taking advantage of these markers, the distinction between melanoma and nevi is possible in most cases, with no need for other ancillary methods such as molecular pathology [1]. PRAME is a cancer testis antigen (CTA, Opa-interacting protein 4) expressed especially in melanomas, among other malignant neoplasms, such as sarcomas, renal cell carcinoma, non-small cell lung cancer and ovarian cancer [8,9,10,11]. A relevant biological implication has PRAME on germ cell tumors of the testis, regulating the “reprogramming” of seminoma cells toward other non-seminomatous germ cell tumors [12]. Further, PRAME is involved in many cellular processes that are relevant for tumor genesis, such as proliferation, anti-apoptosis and metastasis formation [12]. It is only expressed at low levels in reproductive organs, making PRAME an interesting antigen for cell-based immunotherapies [13]. In previous studies, PRAME has been identified as a useful biomarker and has received increased attention as an additional tool in the diagnosis of melanoma [14,15,16]. However, the interpretation of PRAME immunoreactivity is neither clear-cut nor standardized and may therefore bear some pitfalls [17]. Thus, a standardized assessment for this marker is highly desirable. In this study, we use a standardized computer-assisted assessment of PRAME staining to distinguish between thin superficial spreading melanomas (SSM) and DN.

## 2. Results

Overall, 81 PRAME stains of 81 different specimens were analyzed. They included 40 (49.4%) SSM and 41 (50.6%) DN. In the evaluation cohort, 90.6% of the DN were junctional nevi and 9.4% were compound nevi. The mean tumor thickness of the SSM was 0.42 mm (SD 0.2 mm). In the validation cohort, 44.4% of the DN were junctional nevi and 55.6% were compound nevi. The mean tumor thickness of the SSM was 0.66 mm (SD 0.39 mm). Additional clinical features are summarized in Table 1. In the evaluation cohort (*n* = 64), the mean of PRAME-positive cells/mm^2^ in the entire lesion was 455.3 (SD 428.2) in SSM and 60.5 (SD 130.1) in DN. With restriction to the epidermis, the mean was 656.3 (SD 612.9) in SSM and 70.3 (SD 156.8) in DN. The difference in PRAME-positive cells/mm^2^ between SSM and DN was highly significant (*p* < 0.001; Table 2). The ROC AUC of PRAME-positive cells in the entire lesions was 0.866 and showed no difference between the different intensity thresholds (level 1: 0.873, Figure 1, Appendix A). The ROC AUC of PRAME-positive cells in the epidermis was 0.901 for baseline and 0.9 for level 1 (Figure 2). The calculated cut-off score with the highest sum of sensitivity and specificity of 1.594 for the entire lesion was 97.67 cells/mm^2^ (Appendix A). This cut-off value was in line with the classifier evaluation metrics of SPSS, with a maximal Kolmogorow–Smirnow-Test value of 0.594. In the validation cohort (*n* = 17), 16 of 17 cases were correctly classified through the predicted cut-off score (94.1%; Table 3). The mean of PRAME-positive cells per mm^2^ in SSM (*n* = 8) was 1106.8 (SD 843.5) for the entire lesion and 13.9 (SD 22, *n* = 9; *p* < 0.001) for DN.

## 3. Discussion

CTAs are highly expressed in cancer cells and in non-cancer tissues restricted to the reproductive organs. In particular, PRAME is involved in many cellular processes that are relevant for tumor genesis, such as proliferation, anti-apoptosis and metastasis. Thus, PRAME seems to be a suitable target for the differentiation between benign and malignant melanocytic lesions of the skin. Its value in the diagnosis of neoplastic skin lesions has been evaluated for many different entities, such as melanocytic lesions of the nail [18], spitzoid neoplasms [15], desmoplastic melanoma [19], halo nevi [20], nevus-associated melanomas [21], metastatic melanoma [22], and uveal melanoma [23]. Googe et al. reported a focal PRAME expression in 22% of benign nevi (29/134), whereas 80% of melanomas (95/119) were diffusely PRAME-positive [24]. Of note, PRAME has been shown to be a sensitive and specific marker for the distinction of lentigo maligna from melanocytic hyperplasia in chronically sun-damaged skin [25,26]. Furthermore, Olds et al. microscopically distinguished early melanoma from benign pigmented lesions by a manual score of a board-certified dermatopathologist and suggested 10 PRAME-positive cells/mm^2^ as an acceptable threshold of PRAME positivity [27]. However, these studies only focused on healthy or sun-damaged tissue and did not include melanocytic dysplastic (still benign) lesions. To date, only one study compared superficial DN with melanomas by a manual score, revealing PRAME expression in only 1/35 DN (2.9%), while an early-stage melanoma could not entirely be excluded in this single case [17]. Gassenmaier et al. suggested that a high PRAME score (>75% epidermal and >75% dermal melanocytes) should be a threshold for a potential melanoma diagnosis [17]. In order to reach a higher diagnostic accuracy, a double-staining technique (for example, Melan A/PRAME or HMB45/PRAME) can be applied [28].

In this study, we assessed PRAME expression in early invasive SSM and DN and went beyond these previous analyses using a standardized computer-assisted procedure to minimize intra- and inter-observer variability [29]. As DN can have some degree of PRAME reactivity, a clear distinction is even more difficult to determine, and standardized measures are warranted. We calculated a cut-off score of 97.67 cells/mm^2^ for melanoma, with a sensitivity of 71.9% and specificity of 87.5%. The cut-off of 10 PRAME-positive cells/mm^2^, as previously suggested by Olds et al. [27], would equal a sensitivity of 96.4% and a specificity of only 43.7%, according to the coordinates of our ROC, implying that many dysplastic but still benign lesions would have been categorized as melanomas. According to our data, a high PRAME expression of more than 100 cells/mm^2^ is a strong indicator for melanoma, with a specificity of >87%, demonstrating that a high PRAME expression in DN occurs rarely. According to the data and as we could observe in our validation cohort, a high PRAME expression is strongly suggestive of melanoma, although melanoma cannot be fully excluded when PRAME expression is low. Considering the involvement of PRAME in tumorigenic processes, we suggest the use of a cut-off value of 100 cells/mm^2^ for melanoma diagnosis. Furthermore, we did not observe a difference in the ROC AUC between all (baseline) and strong (level 1) PRAME immunoreactive nuclei, indicating that a general immunoreactivity may point to a malignant potential as well as an intense expression of fewer cells, which is in line with a comparably low sensitivity of 71.9% at the cut-off score of 97.67 cells/mm^2^. Additionally, we tried to evaluate the difference between PRAME expression in the epidermis and the entire lesion. Research was published stating that PRAME stains the intraepidermal component in some nevi with a decreasing gradient towards depth [30]. This observation is in line with our results, as the cell density of PRAME-positive cells was higher in the epidermis than in the entire lesion. In both DN and SSM (epidermal and entire lesion), a significant difference in PRAME expression was observed, and the ROC AUC was higher for the epidermis, demonstrating that intraepidermal expression may have a higher significance in the context of the classification of melanocytic lesions.

Limitations of this study were the small sample size and the fact that we only included cases in which PRAME was stained as part of the routine histologic diagnosis. Therefore, on the one hand, PRAME expression might have influenced the diagnosis, resulting in a bias. On the other hand, it demonstrates that exclusively borderline lesions were evaluated, which might have biased a clear cut-off between the two cohorts. Finally, the blinding was impaired, as the investigator could recognize the diagnosis of the samples due to the overall morphology.

In conclusion, PRAME immunostaining seems to be a reliable tool to distinguish between DN and SSM, but PRAME expression is on a continuum, and a clear cut-off is difficult to define. Based on our results, a PRAME expression of more than 100 cells/mm^2^ should raise a high degree of suspicion for melanoma but not exclude a diagnosis of melanoma in lesions with lower expression levels or an exclusively epidermal expression. Lesions with a moderate PRAME expression and indifferent morphologic features will remain a challenge for dermatopathologists.

## 4. Materials and Methods

This study was conducted under approval of an independent research ethics committee of the Friedrich-Alexander-University Erlangen-Nuremberg (approval number 22-368-Br). This retrospective analysis was based on 81 cases, in which PRAME was routinely stained for the classification of SSM and DN between January and August 2022 for the evaluation cohort (*n* = 64) and between September 2022 and March 2023 for the validation cohort (*n* = 17). The histological slides were obtained from the archives of the Unit for Dermatohistology, Department of Dermatology, Uniklinikum Erlangen, Germany. Prior to analysis, the slides were reviewed to exclude bleached and artifact-rich specimens. Patients were sampled and information obtained from the clinical and pathologic records.

### 4.1. Immunohistochemistry

IHC was performed in the certified laboratory of the Unit for Dermatohistology, Department of Dermatology, Uniklinikum Erlangen, Germany, using a fully automated IHC slide staining instrument (BenchMark XT by Roche Diagnostics, Rotkreuz, Switzerland). A commercially available antibody to PRAME (clone QR005; quartett GmbH, Berlin, Germany) with a dilution of 1:300 and an incubation time of 36 min was used. Further, Fast Red chromogen for immunohistochemical staining was applied to rule out interference with unspecific cytoplasmic melanin pigment.

### 4.2. Image Data Acquisition

All slides were digitized with the NanoZoomer-SQ (Hamamatsu Photonics K. K.; Herrsching, Germany) at a 40× magnification, with a resolution of 0.23 µm/pixel. The PRAME-stained slides were analyzed with the bioimage analysis software QuPath (version 0.3.2) [31]. The analysis was performed by one investigator (E.A.T.K.) without knowledge of the histological diagnosis. In QuPath, annotations of the whole lesion (epidermal and dermal) were performed, as well as of the epidermis exclusively. For the detection of PRAME-positive cells, a threshold was established, which identified all cells in the annotated area and distinguished between PRAME-positives and -negatives. Two different intensity thresholds for the parameters of the nucleus (fast-red optical-density mean) were established. One (baseline) that detected all PRAME-positive cells and another one (level 1) that only detected cells with a strong PRAME expression. In summary, the higher the threshold was set, the stronger the intensity of the fast-red-positive cells (PRAME) had to be in order to be recognized as positive. The score of positive cells/mm^2^ for all annotations in different intensity grades was extracted.

### 4.3. Statistical Analysis

The evaluation cohort was split according to the diagnosis (SSM versus DN) and compared using the t-test. A two-sided *p*-value of <0.05 was used for statistical significance. Additionally, the evaluation cohort was classified through a receiver-operating-characteristic curve (ROC-curve) and the area under the curve (AUC). A ROC AUC score of >0.8 was considered good, and >0.9 was considered to be a very good result [32]. In a next step, we calculated a cut-off score through the threshold in the ROC curve with the highest value for the sum of the sensitivity and specificity and additionally with the classifier evaluation metrics of SPSS (including the Kolmogorov–Smirnov test and Gini index). With the cut-off score, the diagnosis of the validation cohort was predicted and verified on the basis of the original diagnosis. All statistical analyses were performed using IBM^®^ SPSS Statistics (version 28.0.0.0, 190; Armonk, NY, USA).

## Figures and Tables

**Figure 1 ijms-24-06388-f001:**
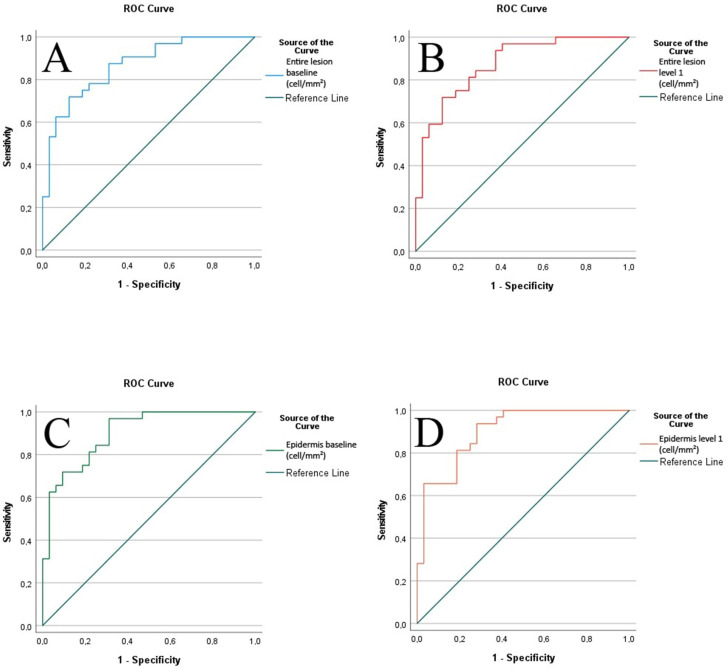
Area under the curve (ROC AUC) of the entire lesion and the epidermis with both thresholds. (**A**) ROC AUC of the entire lesion with baseline threshold (AUC 0.866, CI 95% 0.780–0.953). (**B**) ROC AUC of the entire lesion with level 1 threshold (AUC 0.873, CI 95% 0.789–0.957). (**C**) ROC AUC of the epidermis with baseline threshold (AUC 0.901, CI 95% 0.829–0.973). (**D**) ROC AUC of the epidermis with level 1 threshold (AUC 0.900, CI 95% 0.827–0.973).

**Figure 2 ijms-24-06388-f002:**
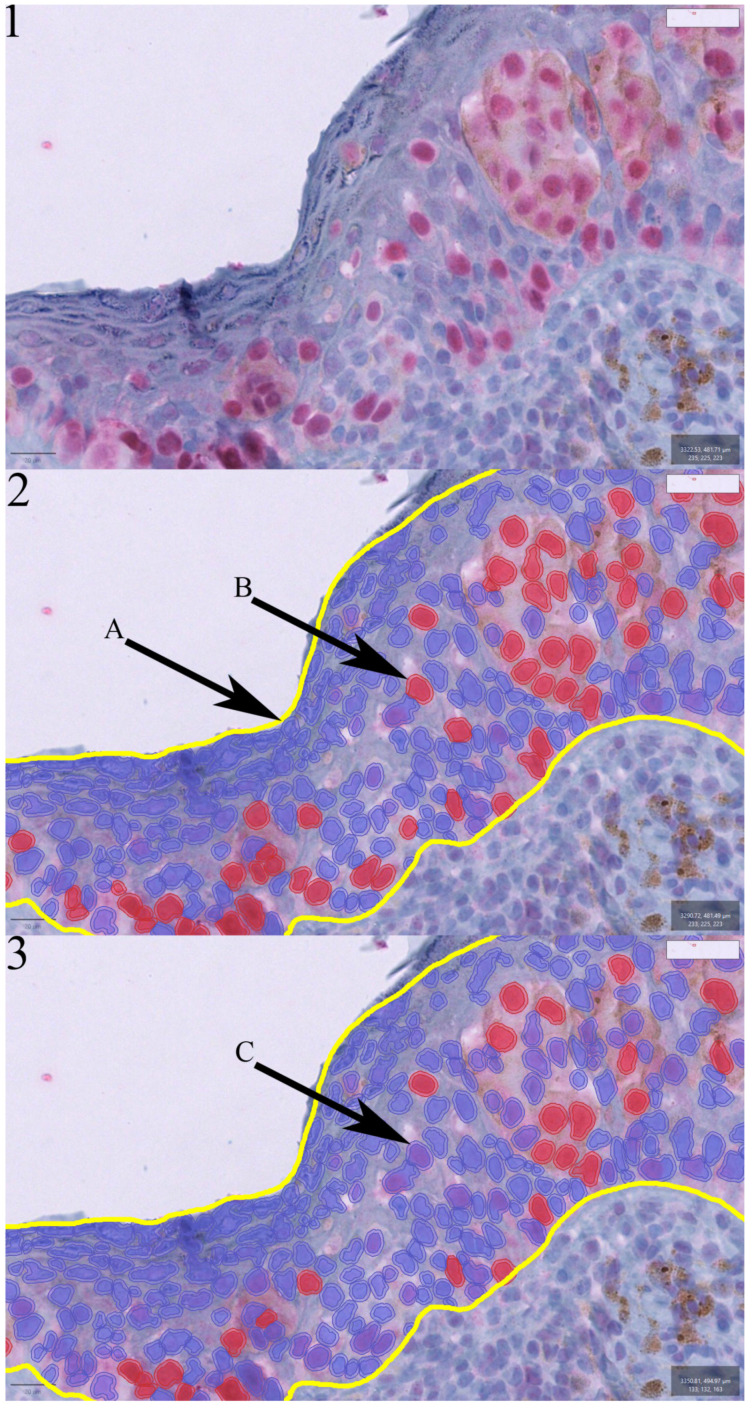
Picture 1 shows an area of a superficial spreading melanoma without annotation. Pictures 2 and 3 present the detection of cells in a given annotated area (2A: yellow frame, epidermal annotation), distinguishing PRAME-positives (2B, red marked cell) and PRAME-negatives (3C, blue marked cell). Two distinct intensity thresholds for the parameters of the nucleus (fast-red optical-density mean) were established. Picture 2 presents the baseline threshold (2B: the cell is detected as positive), and picture 3 demonstrates the level 1 threshold (3C: recognizes the previous positive cell (2B) as negative).

**Table 1 ijms-24-06388-t001:** Characteristics of the study population.

**Evaluation cohort**	**Superficial spreading melanoma**		**Dysplastic nevus**	
N	32		32	
Tumor thickness	Mean: 0.42 mm			

	Standard deviation: 0.2 mm			

Nevus type			Junctional nevus: 90.6% (29/32)	

			Compound nevus: 9.4% (3/32)	

Location (*n*)	Head/Neck: 3		Head/Neck: 3	
	Trunk: 14		Trunk: 18	
	Upper extremities: 10		Upper extremities: 6	
	Lower extremities: 5		Lower extremities: 5	
Gender (*n*)	Female: 18		Female: 18	
	Male: 14		Male: 14	
Age	Mean: 57.56 years		Mean: 54.16 years	
	Standard deviation: 14.2		Standard deviation: 20.1	
**Validation cohort**	**Superficial spreading melanoma**		**Dysplastic nevus**	
N	8		9	
Tumor thickness	Mean: 0.66 mm			

	Standard deviation: 0.39 mm			

Nevus type			Junctional nevus: 44.4% (4/9)	

			Compound nevus: 55.6% (5/9)	

Location	Head/Neck: 1		Head/Neck: 0	
	Trunk: 3		Trunk: 6	
	Upper extremities: 0		Upper extremities: 2	
	Lower extremities: 4		Lower extremities: 1	
Gender (*n*)	Female: 4		Female: 2	
	Male: 4		Male: 7	
Age	Mean: 57.75 years		Mean: 52.9 years	
	Standard deviation: 22.1		Standard deviation: 22.4	

**Table 2 ijms-24-06388-t002:** Comparison of dysplastic nevi (DN) and superficial spreading melanomas (SSM) according to the PRAME-positive cells/mm^2^ in the evaluation cohort.

Evaluation Cohort	Diagnosis	N	Mean	Standard Deviation	Standard Error of the Mean	*t*-Test (Two-Sided)
Entire lesion (positive cells/mm^2^)	DN	32	60.5	130.1	23	*p* < 0.001
	SMM	32	455.3	428.2	75.6	
Epidermis (positive cells/mm^2^)	DN	32	70.3	156.8	27.7	*p* < 0.001
	SMM	32	656.3	612.9	108.3	

**Table 3 ijms-24-06388-t003:** Prediction of the validation cohort according to the cut-off value of 97.7 positive cells/mm^2^.

Validation Cohort	Positive Cells/mm^2^—Entire Lesion	Expected Coordinate in ROC: Sensitivity	Expected Coordinate in ROC: 1-Specificity	Prediction According to Cut-Off Value (K-S-Test) of 97.7 Positive Cells/mm^2^	Correct Prediction	Accuracy
Sample 1	413.73	0.438	0.031	Melanoma	Yes	94.1%
Sample 2	0	1.000	1.000	Nevus	Yes	
Sample 3	1406.9	0.000	0.000	Melanoma	Yes	
Sample 4	131.17	0.625	0.063	Melanoma	Yes	
Sample 5	0	1.000	1.000	Nevus	Yes	
Sample 6	1747	0.000	0.000	Melanoma	Yes	
Sample 7	0	1.000	1.000	Nevus	Yes	
Sample 8	1959.5	0.000	0.000	Melanoma	Yes	
Sample 9	2219.6	0.000	0.000	Melanoma	Yes	
Sample 10	0	1.000	1.000	Nevus	Yes	
Sample 11	57.13	0.781	0.313	Nevus	Yes	
Sample 12	0	1.000	1.000	Nevus	Yes	
Sample 13	89.49	0.750	0.219	Nevus	No	
Sample 14	887	0.188	0.000	Melanoma	Yes	
Sample 15	31.42	0.906	0.406	Nevus	Yes	
Sample 16	0	1.000	1.000	Nevus	Yes	
Sample 17	37.1	0.875	0.375	Nevus	Yes	

## Data Availability

Data are contained within the article or Appendix A.

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
