# Peer review of "Standardized Computer-Assisted Analysis of PRAME Immunoreactivity in Dysplastic Nevi and Superficial Spreading Melanomas"

_ijms, 2023, doi:10.3390/ijms24076388_

Round 1

Reviewer 1 Report

The paper is dedicated to the interesting problem – how easily differentiate the melanoma from unmalignizated nevi. The author developed new mathematical approach for the analysis of the immunocytochemical identification of melanoma on the base of mathematical methods.

The authors point out that the interpretation of the intensity and distribution of PRAME immunostaining makes interpretation difficult. So they developed the mathematical way to interpret the ICC staining, and if so, it is a big methodological value of this work.

In this study, a standardized computer-assisted assessment of PRAME staining was developed to distinguish between thin superficial spreading melanomas and dysplastic nevi.

However, I with my experience – do not understand the figure 2 at all, because: 1) the insets in the upper right corners are too small to see them; 2) the left and right columns seems absolutely identical on the first view; 3) although the quality of the micrographs is very good! there are no any indications what we need to see? what we need to compare? any arrows, arrowheads, letters?? and these yellow frame is so thin that could not be distinguished by eye (nearly).

 In my opinion it should be more clear for reader because all your mathematical approach is based on these ICC images!  Probably make more (two) photos with higher magnification (except these) and please insert some markers in your figures.

English is fine.

 The Introduction is conscious and clear. Discussion and Methods section is clear.

Tables seems clear to me, and Supplementary materials, although they concern of statistical and mathematical analysis of the results obtained with immunocytochemistry.

Author Response

Comment #1:

The paper is dedicated to the interesting problem – how easily differentiate the melanoma from unmalignizated nevi. The author developed new mathematical approach for the analysis of the immunocytochemical identification of melanoma on the base of mathematical methods.

The authors point out that the interpretation of the intensity and distribution of PRAME immunostaining makes interpretation difficult. So they developed the mathematical way to interpret the ICC staining, and if so, it is a big methodological value of this work.

In this study, a standardized computer-assisted assessment of PRAME staining was developed to distinguish between thin superficial spreading melanomas and dysplastic nevi.

However, I with my experience – do not understand the figure 2 at all, because: 1) the insets in the upper right corners are too small to see them; 2) the left and right columns seems absolutely identical on the first view; 3) although the quality of the micrographs is very good! there are no any indications what we need to see? what we need to compare? any arrows, arrowheads, letters?? and these yellow frame is so thin that could not be distinguished by eye (nearly).

In my opinion it should be more clear for reader because all your mathematical approach is based on these ICC images!  Probably make more (two) photos with higher magnification (except these) and please insert some markers in your figures.

Response #1:

We thank the reviewer for this comment and modified figure 2 for better understanding.

Comment #2:

English is fine.

The Introduction is conscious and clear. Discussion and Methods section is clear.

Tables seems clear to me, and Supplementary materials, although they concern of statistical and mathematical analysis of the results obtained with immunocytochemistry.

Response #2:

We thank the reviewer for his positive feedback.

Reviewer 2 Report

The manuscript presented by Heppt and colleagues is based on a computer-aided assessment of PRAME immunostaining in superficial spreading melanoma and age and gender matched displastic nevi.  Overall, the originality of the study is limited to the application of a computer assessment of a well known marker for melanoma. In my opinion, as stated also by the authors, the study has several limits: the small sample size and the analyis limited to border lesions. Furthermore, the application of the computer assessment continues to be inconclusive in case  of melanoma with low/moderate  PRAME expression. I believe that in the present form the manuscript does not  reach a publication level. An extensive revision is needed implementing the study. Authors should analyze another test set of samples to validate their cut-off for the analysis. The validation set should include at least the same number of samples included in the training set. It would be also useful to test the performarce of their cut-offs in other histological type of melanomas and benign pigmented lesions.

Minor concerns:

1. It is not clear to me the level 1 threshold for PRAME immunoistaining. What does it mean strong intensity? 2+, 3+? What is the level of reproducibility of the abovementioned threshold? And of the cut-off?

Author Response

Comment #1: The manuscript presented by Heppt and colleagues is based on a computer-aided assessment of PRAME immunostaining in superficial spreading melanoma and age and gender matched displastic nevi.  Overall, the originality of the study is limited to the application of a computer assessment of a well known marker for melanoma. In my opinion, as stated also by the authors, the study has several limits: the small sample size and the analyis limited to border lesions. Furthermore, the application of the computer assessment continues to be inconclusive in case  of melanoma with low/moderate  PRAME expression. I believe that in the present form the manuscript does not  reach a publication level. An extensive revision is needed implementing the study. Authors should analyze another test set of samples to validate their cut-off for the analysis. The validation set should include at least the same number of samples included in the training set. It would be also useful to test the performarce of their cut-offs in other histological type of melanomas and benign pigmented lesions.

Minor concerns:

  1. It is not clear to me the level 1 threshold for PRAME immunoistaining. What does it mean strong intensity? 2+, 3+? What is the level of reproducibility of the abovementioned threshold? And of the cut-off?

Response #1:

We thank the reviewer for his feedback. The intensity of a stain means the strength of the expression of the color “fast red”, which correlates with the PRAME expression of that cell. We wanted to understand the expression and their diagnostic relevance. With the implementation of a threshold, we investigated if a strong expression of PRAME correlates with a malign diagnoses and compare it to the cells with at least an average expression (baseline). Definitely, the setting of the threshold is in the judgement of the investigator and therefore subjective to a certain degree, but within the generally accepted principles. According to this the cut-off score and the threshold should be reproducible.

Reviewer 3 Report

A very interesting and necessary study The number of investigated cases seems insufficient for statistical evaluation. It is unclear what the authors consider a dysplastic nevus - junctional nevus - compound nevus are in themselves something different from a dysplastic nevus - an appropriate mention, for example, of a dysplastic nevus with a junctional type - a size greater than 6 mm... Nevertheless, the work is exciting and should be expanded or designated as a pilot study...

Author Response

Comment #1: A very interesting and necessary study The number of investigated cases seems insufficient for statistical evaluation. It is unclear what the authors consider a dysplastic nevus - junctional nevus - compound nevus are in themselves something different from a dysplastic nevus - an appropriate mention, for example, of a dysplastic nevus with a junctional type - a size greater than 6 mm... Nevertheless, the work is exciting and should be expanded or designated as a pilot study...

Response #1:

We thank the reviewer for his feedback and would like to provide clarification. In general, we distinguish a common melanocytic nevus between junctional-, compound-, and dermal-nevus. Each of these nevi can be dysplastic depending on its morpholic features. You may also call it e.g. a junctional nevus with dysplastic features. 

Round 2

Reviewer 2 Report

The authors did not make the modifications I required, notably the test of another set of patients. I appreciate before publication that they provide with this additional test.

Regarding the answear on the intensity of the staining, there is maybe a misunderstanding. It is clear to me what the intesity means in IHC measure. However, the figure 2 is a response for the question I posed.

Regarding figure 2, it would be helpful to subdivide the image in panels (A)……instead of recalling in the legend lower and higher pictures.

Author Response

We thank the reviewer for his feedback. The valuable feedback has significantly improved the manuscript. We feel that we have addressed all specific concerns.

Reviewer 3 Report

The authors significantly improved the manuscript. This is an interesting work that can be developed in the future.  

Author Response

We thank the reviewer for his positive feedback.
